# Evaluation of DNA Methylation Profiles of LINE-1, Alu and Ribosomal DNA Repeats in Human Cell Lines Exposed to Radiofrequency Radiation

**DOI:** 10.3390/ijms24119380

**Published:** 2023-05-27

**Authors:** Francesco Ravaioli, Maria Giulia Bacalini, Cristina Giuliani, Camilla Pellegrini, Chiara D’Silva, Sara De Fanti, Chiara Pirazzini, Gianfranco Giorgi, Brunella Del Re

**Affiliations:** 1IRCCS Istituto Delle Scienze Neurologiche di Bologna, 40139 Bologna, Italy; francesco.ravaioli@ausl.bologna.it (F.R.); mariagiulia.bacalini@isnb.it (M.G.B.); camilla.pellegrini@ausl.bologna.it (C.P.); chiara.dsilva@ausl.bologna.it (C.D.); sara.defanti@ausl.bologna.it (S.D.F.); 2Laboratory of Molecular Anthropology and Centre for Genome Biology, Department of Biological, Geological and Environmental Sciences (BIGEA), University of Bologna, 40126 Bologna, Italy; cristina.giuliani2@unibo.it; 3Department of Medical and Surgical Sciences (DIMEC), University of Bologna, 40126 Bologna, Italy; chiara.pirazzini5@unibo.it; 4Department of Pharmacy and Biotechnology (FABIT), University of Bologna, 40126 Bologna, Italy; gianfranco.giorgi@unibo.it

**Keywords:** radiofrequency electromagnetic field, DNA methylation, neuroblastoma, LINE-1, Alu, ribosomal DNA

## Abstract

A large body of evidence indicates that environmental agents can induce alterations in DNA methylation (DNAm) profiles. Radiofrequency electromagnetic fields (RF-EMFs) are radiations emitted by everyday devices, which have been classified as “possibly carcinogenic”; however, their biological effects are unclear. As aberrant DNAm of genomic repetitive elements (REs) may promote genomic instability, here, we sought to determine whether exposure to RF-EMFs could affect DNAm of different classes of REs, such as long interspersed nuclear elements-1 (LINE-1), Alu short interspersed nuclear elements and ribosomal repeats. To this purpose, we analysed DNAm profiles of cervical cancer and neuroblastoma cell lines (HeLa, BE(2)C and SH-SY5Y) exposed to 900 MHz GSM-modulated RF-EMF through an Illumina-based targeted deep bisulfite sequencing approach. Our findings showed that radiofrequency exposure did not affect the DNAm of Alu elements in any of the cell lines analysed. Conversely, it influenced DNAm of LINE-1 and ribosomal repeats in terms of both average profiles and organisation of methylated and unmethylated CpG sites, in different ways in each of the three cell lines studied.

## 1. Introduction

In 2011, radiofrequency electromagnetic fields (RF-EMFs), ranging from 30 kHz to 300 GHz of the electromagnetic spectrum, were classified by the International Agency for Research on Cancer (IARC) as possibly carcinogenic to humans (risk group 2B) [1]. This type of radiation is emitted by everyday devices including mobile phones, radios, televisions, medical equipment and by many other sources in occupational and general environmental settings. For mobile devices, the specific absorption rate (SAR) of radiofrequency radiation emitted during usage varies depending on the model, the distance and the quality of the connection with the base-station antenna [1]. There is a growing concern about possible harmful health effects, since human exposures are ubiquitous and are rapidly increasing in the world.

Genotoxic events are considered to be an initial step in carcinogenesis, so during the last decade, many investigations have been carried out on the possible genotoxic effects of RF-EMFs, as summarised by several reviews [2,3,4,5]. To date, no consistent data have emerged about DNA damage induction, as evaluated through analysis of various endpoints such as induction of chromosomal aberrations, micronuclei, sister chromatid exchange, mutations, single and double DNA strand breaks and DNA degradation. Interestingly, the number of published studies reporting increased genetic damage in cells exposed to RF-EMF is inversely proportional to the number of quality control measures [5], so the inconsistency of the results could be largely due to the poor quality of many studies [6].

Recent evidence indicates that carcinogenesis can also be induced by non-genotoxic agents through epigenetic alterations [7], which can affect gene function, leading to cellular neoplastic transformation without changing the DNA sequence. Therefore, to evaluate the potential carcinogenicity of RF-EMF exposure, possible epigenetic effects should be investigated.

It was reported that the Earth’s magnetic field is involved in DNA methylation (DNAm) regulation [8,9]; therefore, it was reasonable to suspect that exposure to man-made electromagnetic fields, which are greater than natural ones, might interfere with DNAm, causing dysfunction. Accordingly, previous studies reported alterations of DNAm upon exposure to RF-EMF [10,11]. Mokarram et al. showed that 900 MHz RF-EMF altered the DNAm of estrogen receptors in rat colon cells, while Kumar et al. reported a global decrease in DNAm levels in rat hippocampus exposed to RF-EMF. Importantly, in the last study, the effect on DNAm was more pronounced with increasing frequency (900–2450 MHz) and exposure time (one-month to six-month exposure group) [11].

In the present study, we focused on DNAm, since it is the epigenetic mechanism that has been more extensively studied in relation to environmental exposures [12]. In particular, we evaluated DNAm of repetitive DNA elements (RE-DNA), which occupy a large part of the genome and whose deregulation has been implicated in carcinogenesis [13,14].

We focused on three types of RE-DNA: (1) long interspersed nuclear elements-1 (LINE-1), (2) Alu short interspersed nuclear elements (SINEs) and (3) the DNA sequences encoding RNA ribosomal (rDNA).

LINE-1s are the only active retroelements and constitute approximately 17% of the human genome [15]. A full-length LINE-1 element is about 6 kb and is composed of a 5′ untranslated region (5′UTR) containing sense and antisense promoters, two open reading frames (ORF1 and ORF2), encoding proteins involved in the retrotransposition process and a 3′ untranslated region (3′UTR) with a polyadenylation site [16]. After transcription, the LINE-1 retroelement can be inserted into another genomic site, causing genetic instability [17]. The methylation of the CpG island in LINE-1 5′UTRs plays an important role in the repression of LINE-1 transcription and retrotransposition; moreover, it can affect the regulation of the expression of surrounding genes [18,19,20]. Alterations of the LINE-1 5′UTR promoter methylation have been observed in human cancer cells [21,22,23], and they are considered a promising biomarker of cancer development [14]. These alterations can be a consequence of exposure to various environmental stressors [24,25], and it has been proposed that they should be included in the health risk assessment of environmental factors [26].

Alu elements are primate-specific SINE, which comprise about 11% of the genome. They are 300 base pair sequences and are subdivided into three families (J, S and Y) due to different consensus sequences and different times of appearance in the human lineage. Alu elements are nonautonomous retrotransposons; however, they can copy and paste themselves through the enzymatic activity of LINE-1-ORF2 [27]. Methylation represses Alu transcription and activity [28], and aberrant Alu methylation has been found in various kinds of tumours [29]. 

rDNAs are the most abundantly expressed genes in the eukaryotic genome. In the human genome, rDNAs (encoding 18S, 5.8S and 28S rRNA) are tandemly arrayed in a head-to-tail fashion in nucleolar organiser regions (NORs) on the short arms of five pairs of acrocentric autosomal chromosomes (13, 14, 15, 21 and 22) [30]. Each array consists of multiple rDNA repeat units that vary in number among individuals and chromosomes, ranging from 60 to >800, with a mean copy number of 400 in a diploid human genome. Each unit contains a single transcription unit (~13 kb), encoding a 45S rRNA precursor, which is processed to form the 18S, 5.8S and 28S rRNA molecules, and a nontranscribed spacer (~30 kb) (intergenic spacer, IGS) [31]. IGS contains promoters, multiple repetitive sequences and regions producing non-coding RNAs involved in various cellular processes, including stress reactions. A single unit contains more than 1500 CpG sites [32] and shows high levels of methylation and particular distribution of methylation on DNA strands. The methylation of promoters abolishes rDNA transcription by inhibiting the assembly of the transcription initiation complex; indeed, promoters of active rRNA genes are usually hypomethylated and are associated with acetylated histones [33]. Recent evidence indicates that the methylation of the gene body maintains rDNA transcription by preventing the enrichment of repressive histone modifications [34]. It was reported that the methylation profile of DNA sequences encoding ribosomal RNAs (rRNA) plays a role in ageing [32,35] and in cancer onset [36,37].

Some studies have been conducted to ascertain whether the exposure to extremely low-frequency magnetic fields (ELF-EMF) could affect the DNAm of RE-DNA [38,39,40,41], but so far, this issue has been little explored in relation to radiofrequency exposure. 

One limitation of existing studies exploring the impact of the magnetic field and radiofrequency radiation exposure on epigenetic regulation consists in the large heterogeneity in terms of study models and experimental design adopted [41].

The aim of the present study is to assess this endpoint in human cells exposed to 900 MHz GSM-modulated RF-EMF, which is an important frequency for mobile communication electromagnetic radiation emitted from GSM mobile phones and from their base station antennas. To this purpose, DNA methylation profiles of the LINE-1 5′-UTR promoter, Alu sequences and three regions of rDNA (rDNA promoter, 18S and 28S) were investigated in an epithelial human cell line (HeLa) and in two neuroblastoma human cell lines (Be(2)C and SH-SY5Y) exposed to RF-EMF or to sham (control).

All three cell lines employed have been used in several studies as models to assess the biological effects of EMF exposure. and its impact on epigenetic regulation and cellular homeostasis [39,40,42,43,44,45,46,47]. In particular, since it had been suspected that radiofrequency exposure could increase the incidence of brain and nervous system dysfunction [1], we used BE(2)C and SH-SY5Y neuroblastoma cell lines, which show biochemical characteristics of neuron cells [48] and have often been employed as in vitro models of neuronal function and pathology [49,50].

## 2. Results

### 2.1. Validation of the Assay for the Measurement of RE-DNAm

To measure the DNAm of RE-DNA, we used a bisulfite-targeted sequencing assay as described in Materials and Methods. We checked the assay for bisulfite-PCR bias, a phenomenon associated with the preferential amplification of bisulfite-converted DNA according to its original DNAm status. To this aim, we analysed samples at known DNAm percentages (0%, 25%, 50%, 75% and 100%) and then assessed the correlation between observed and expected DNAm values for all CpGs of each RE target (Appendix A). A high correlation between expected and observed DNAm values was observed for all CpGs assessed in the LINE-1 target (Pearson R > 0.94, *p*-value < 0.01) (Appendix A). Similarly, for rDNA targets (namely RiboProm1, RiboProm2, 18S1 and 28S), we observed a highly significant correlation for the vast majority of CpGs assessed (Pearson R > 0.96, *p*-value < 0.01) (Appendix A). The correlation between expected and observed DNAm values for CpGs belonging to the Alu target tended to be lower with respect to the other targets, with some of the CpG sites not reaching statistical significance (Appendix A). CpGs having a low (Pearson R < 0.7) or non-significant correlation (*p*-value > 0.05) were considered not reliable and were excluded from the analyses.

### 2.2. Effects of Electromagnetic Radiation on RE DNAm

To study the effects of radiofrequency exposure on epigenetic regulation of repetitive elements, HeLa, BE(2)C and SH-SY5Y cell lines were exposed to 900 MHz GSM-modulated RF-EMF at a SAR of 1 W/kg. DNAm of LINE-1, Alu and rDNA elements was evaluated.

The analysis of LINE-1 DNAm in the HeLa cell line after radiofrequency exposure showed a significant, albeit small, hypermethylation for CpGs X70 (ΔDNAm:0.019;
*p*-value: 0.033), X183 (ΔDNAm:0.017; *p*-value: 0.022), X190 (ΔDNAm:0.022; *p*-value: 0.007) and X217 (ΔDNAm:0.015; *p*-value: 0.009) compared to sham-exposed samples. In SH-SY5Y and BE2C cell lines, we did not find significant changes in any of the CpGs assayed in the LINE-1 target, although a trend towards hypermethylation was observed in the SH-SY5Y cell line (Figure 1 and Appendix A).

The DNAm profile of the promoter (RiboProm1 and RiboProm2) and of the body (18S1 and 28S) regions of rDNA was affected by radiofrequency exposure in different ways in the three cell lines examined (Figure 2 and Figure 3 and Appendix A).

In the SH-SY5Y cell line, a large group of CpGs, mostly on the 5′ of RiboProm2 target, was found significantly hypermethylated following RF-EMF exposure (ΔDNAm ranging from 0.014 to 0.030; *p*-values ranging from 0.012 to 0.044) (Figure 2B and Appendix A). A larger hypermethylation trend was observed for the amplicon targeting 18S, with CpGs X135 (ΔDNAm:0.079;
*p*-value: 0.041), X179 (ΔDNAm:0.087;
*p*-value: 0.042) and X190 (ΔDNAm:0.070;
*p*-value: 0.042) reaching statistical significance (Figure 3A, Appendix A). On the other hand, no significant DNAm changes were found for both RiboProm1 (Figure 2A and Appendix A) and 28S (Figure 3B and Appendix A).

In the BE(2)C neuroblastoma cell line, we observed hypomethylation on both amplicons targeting the rDNA promoter following RF-EMF exposure. DNAm changes in RiboProm1 amplicon were small but significant for the majority of the target’s CpGs (ΔDNAm ranging from −0.026 to −0.014; *p*-values ranging from 0.004 to 0.045), whereas for RiboProm2, only CpG X159 (ΔDNAm:−0.033; *p*-value: 0.048) reached statistical significance (Figure 2, Appendix A). The exposure of BE(2)C neuroblastoma cells to RF-EMF did not induce any significant DNAm changes for the CpGs belonging to 18S, while it promoted hypomethylation of CpGs at the 3′ end of the 28S amplicon, with CpGs X197 (ΔDNAm: −0.018; *p*-value: 0.009) and X205 (ΔDNAm: −0.014; *p*-value: 0.019) reaching statistical significance (Figure 3 and Appendix A).

In the HeLa cell line, DNAm profiles did not show any significant changes for any rDNA-targeting amplicons with the exception of 28S. Indeed, radiofrequency exposure stimulated hypomethylation of CpGs on the 5′end of the 28S amplicon, with CpGs ×29 (ΔDNAm:−0.021 *p*-value: 0.010), ×72 (ΔDNAm: −0.025; *p*-value: 0.002), ×83 (ΔDNAm:−0.017; *p*-value: 0.049) and ×99 (ΔDNAm:−0.015; *p*-value: 0.031) reaching statistical significance (Figure 3B and Appendix A).

Finally, no DNAm changes were generally found for the CpGs of the amplicons targeting the body of the Alu repeat unit in any of the cell lines assayed with the exception of a slight but significant hypermethylation of CpG X118 (ΔDNAm:−0.005; *p*-value: 0.038) in HeLa cell lines (Figure 4 and Appendix A).

### 2.3. Effects of Electromagnetic Radiation on RE DNAm Epihaplotype Diversity

To further explore the effects elicited by radiofrequency radiation on the DNAm profiles of repetitive elements in the various cell lines assayed, we evaluated DNAm epihaplotype diversity.

The AmplimethProfiler tool has a built-in, qiime-based analytical pipeline that allows one to calculate Shannon diversity indexes via the rarefaction method for each target (Section 4.6 and Section 4.7) Filtering of samples with low coverage (Section 4.6) resulted in the removal of one HeLa cell replicate for all the analysed regions.

HeLa cells showed a small but significant (*p*-value: 0.01) increase in epihaplotype diversity for the LINE-1-targeting amplicon upon exposure to radiofrequency radiation (Figure 5 and Appendix A). Similarly, a small and nearly significant (*p*-value: 0.05) increase in epihaplotype diversity for LINE-1 was observed in radiofrequency-exposed SH-SY5Y neuroblastoma cells (Figure 5 and Appendix A).

For rDNA-targeting amplicons, we found a significant decrease in the epihaplotype diversity index for amplicons targeting the proximal rDNA promoter RiboProm2 (*p*-value: 0.023) (Figure 6 and Appendix A) and 28S (*p*-value: 0.025) (Figure 7 and Appendix A) upon radiofrequency exposure of the BE(2) neuroblastoma cell line. We did not find any significant changes in epihaplotype diversity in SH-SY5Y and HeLa cell lines.

Finally, for Alu-targeting amplicons, no significant differences in epihaplotype diversity indexes were found in any of the cell lines assayed (Figure 8).

## 3. Discussion

In this work, we evaluated the impact of RF-EMF exposure on the DNAm of LINE-1, Alu and rDNA repetitive elements in human neuroblastoma and cervical cancer cell lines. The effects on DNAm were evaluated in terms of both alterations in the average DNAm level and alterations of DNAm profile (epihaplotype) richness and distribution.

Our data indicate that RF-EMF exposure can induce alterations in the average DNAm level of LINE-1 CpGs (limited to the HeLa cell line) and of rDNA (for all three cell lines tested). No effect was instead found for the methylation of CpGs of the Alu repeat in any of the cell lines. Previously, Alu methylation was evaluated by Benassi et al. [40] in SH-SY5Y cells exposed to ELF-EMF, and no effect was found, similar to our observation.

With regard to LINE-1, we observed an increase in the DNAm of some CpG in HeLa cells; this finding could be consistent with the observation, reported in a previous paper [43], of a significant reduction in the mRNA levels of LINE-1 in HeLa cells exposed to 900 MHz. Indeed, increased DNAm is usually associated with decreased RNA transcription.

Differently, no effect was found on LINE-1 methylation in neuroblastoma cells (SH-SY5Y and BE(2)C cells). Similar investigations were previously carried out in SH-SY5Y and BE(2)C cells exposed to ELF-EMF (50 Hz, 1 mT) in two different studies. In agreement with our results, although concerning a different type of electromagnetic field, no significant DNAm changes were found either in exposed BE(2)C [39] or in exposed SH-SY5Y cells [40].

With regard to rDNA methylation, alterations were observed in all three cell lines: exposed HeLa cells showed hypomethylation of some CpG on the 3′ end of the 28S amplicon target and exposed SH-SY5Y cells showed hypermethylation of the 5′ of RiboProm2 and 18S targets, while exposed BE(2)C neuroblastoma cells showed hypomethylation of the 5′-end of RiboProm1, the 5′-end of RiboProm2 and the 3′-end of 28S amplicon targets. It is well known that rDNA methylation patterns contribute to the regulation of the expression of ribosomal RNA (rRNA) related to the translational rate, which depends on cell-specific homeostasis requirements [51]. Therefore, it is not surprising that the three cell lines analysed showed different changes in rDNA methylation in response to RF-EMF exposure.

To date, DNAm of rDNA has never been analysed in exposed cells to ELF-EMF or to RF-EMF, but it has been investigated in both physiological and pathological conditions including studies focused on cancer [36,37], neurodegenerative diseases [52], ageing and ageing-related conditions [32,35]. For instance, changes in rDNA methylation have been described in human blood cells and hepatocytes with ageing [53], in blood samples from persons with Down Syndrome [54] and in brain samples from patients with Alzheimer’s disease [52,55].

In addition to assessing the impact of radiofrequencies on the average DNAm level, we evaluated their possible effects on the organisation of methylated and unmethylted CpG sites through the analysis of epihaplotype richness and distribution.

As far as we know, the analysis of DNAm epihaplotypes, by calculating the Shannon diversity index, has never been performed in cells that are EMF-exposed. Our data indicate that RF-EMF exposure can induce changes in the DNAm epihaplotype diversity. In some cases, such as for LINE-1 DNAm in HeLa and SH-SY5Y cells, epihaplotype diversity increased in exposed samples, whereas in other cases, such as rDNA DNAm in BE(2)C cells, it decreased. Changes in the distribution and richness of epihaplotypes may indicate a dysfunction in the maintenance of epigenetic patterns, which can contribute to epigenetic instability. Changes in the Shannon diversity index have been found in ageing [56,57] and in progeroid syndromes such as Down Syndrome [54].

As a whole, our data indicate that RF-EMF exposure affects the DNAm of tested RE and might induce epigenetic instability. These data add further evidence to our knowledge of the effects of RF-EMF on DNAm [10,11]. Our findings pertain to the exposure to 900 MHz; however, effects on DNAm have been found using both frequencies higher than 900 Hz [11] and frequencies lower than 900 MHz, such as ELF-MF (50 Hz) [41].

A weakness of the present study is the limited number of analysed samples and that the use of cell lines may not exactly reflect what happens in vivo. Analysis of primary tissues and stem cells would provide further knowledge.

The issue of what molecular mechanisms underlie the biological effects of EMFs on DNAm regulation is still an open question [58,59,60]. One intriguing hypothesis is that the biological effects of EMFs are mediated by free radicals. Emerging evidence suggests EMFs can induce changes in the energy levels of certain molecules, affecting the concentration of free radicals such as reactive oxygen species (ROS) [44,61,62,63,64,65,66,67,68]. In turn, ROS can affect both genome-wide and site-specific DNAm patterns [69,70,71,72,73]. In this study, this possible biological mechanism of RF-EMF was not evaluated in depth, as ROS levels were not measured, nor were ROS-exposed controls included. Future studies will be needed to determine whether ROS production may play a role in the alterations of DNAm of RE in the context of radiofrequency radiation exposure.

Moreover, our results indicate that the effect on DNAm of LINE-1 and rDNA strongly depends on the cellular system considered and on the genetic background, in agreement with the fact that epigenetic modifications are tissue-specific and show inter-individual variability.

Previous studies showed that the biological model is relevant in EMF exposure studies. For example, Martin et al. [74] studied the expression of three genes in three primary cultures of human keratinocytes subjected to identical exposure conditions (60.4 GHz) and found 3 different expression profiles. They speculated that this fact could be due to epigenetic modifications specific to each cellular model used. Similar results were found by Del Re et al. [43] studying the expression of some repetitive elements in HeLa, Be(2)C and SH-SY5Y cell lines.

Collectively, these data indicate that many different models should be studied to understand the epigenetic effects of EMF exposure on biological samples.

In conclusion, our results suggest that RF-EMF exposure can affect the DNAm of LINE-1 and rDNA sequences, which are highly represented in the human genome, and that the effects depend on cell context (cell genome and cell type). Further investigations on the epigenetic effects of RF-EMF would be useful both for a better understanding of RF-EMF risks and safety assessment.

## 4. Materials and Methods

### 4.1. Exposure System

The exposure system was kindly loaned to us by Prof. Laura Calzà (University of Bologna). The exposure system and the dosimetry were previously described in great detail [75,76]. Briefly, the system is composed of two twin “Transverse Electro Magnetic (TEM) cells” with two independent power supply chains (two GSM signal generators and two EMF amplifiers) to obtain sham and exposed samples at the same time, and it was positioned inside a Heraeus incubator (B-5060, Heraeus, Hanau, Germany) (37 °C, 5% CO_2_). Each TEM cell has a squared cross-section (14 cm wide) and can contain 6 4-multiwell plates (66 mm × 66 mm external size). The RF-EMF (900 MHz; GSM basic with pulsed modulation at 217 Hz) is uniformly absorbed in the two wells of the two symmetric plates closer to the inner copper septum of the TEM cell, resulting in a SAR of 1 W/Kg. Therefore, only the septum-adjacent wells were used in the experiments in accordance with Del Vecchio et al. [75,76]. The system was controlled by LabVIEW 6.1 software (National Instruments, Austin, TX, USA). Experiments were conducted in blind modality. In each assay, cells were plated and allowed to adhere overnight and then exposed to 900 MHz GSM-modulated RF-EMF at SAR of 1 W/kg or to sham. After 48 h of exposure, DNA was extracted from each sample.

### 4.2. Cell Culture

HeLa cells, neuroblastoma BE(2)C cells and neuroblastoma SH-SY5Y cells were kindly provided by Prof. Perini (University of Bologna, Italy). All cell lines were grown in DMEM medium (EuroClone, Milano, Italy), containing 10% heat-inactivated fetal bovine serum (EuroClone, Milano, Italy), 100 UI/mL penicillin (Sigma, Ronkonkoma, NY, USA) and 100 μg/mL streptomycin (Sigma), under a 5% CO_2_ humified atmosphere at 37 °C. In each assay, cells were plated in 4-well plates (15 mm diameter; 1 mL/well; culture area 1.9 cm^2^) (Thermo Scientific Nunc, A/S Roskilde, Denmark) at a density of 80,000 cells/well and allowed to adhere overnight, and then the plates were randomly assigned to each transverse electromagnetic (TEM) chamber and exposed to 900 MHz GSM-modulated RF-EMF at SAR of 1 W/kg or to sham.

### 4.3. DNA Extraction and Bisulfite Conversion

Genomic DNA was extracted by using the QIAmp DNA Mini Kit (QIAgen, Hilden, Germany) according to the manufacturer’s instructions and quantified using the Qubit dsDNA BR Assay Kit (Thermo Fisher Scientific, Waltham, MA, USA). Then, 500 ng was treated with sodium bisulfite using the EZ DNA Methylation Direct Kit (Zymo Research, Irvine, CA, USA) as indicated by the manufacturer. Each sample was sequenced in triplicate. DNA methylation calibration curves were prepared by combining Universal Methylated and Universal Unmethylated DNA samples (Millipore, Burlington, MA, USA) to generate standards at 0, 25, 50, 75 and 100% DNAm levels. Each point of the curve was sequenced in triplicate

### 4.4. Library Preparation and Bisulfite-Targeted Sequencing

DNAm of RE-DNA was analysed using a targeted bisulfite sequencing approach. Primers for rDNA distal (RiboProm1) and proximal (RiboProm2) promoters as well as for 5′UTR of LINE-1 were previously published [77], whereas primers targeting 18S, 28S and Alu regions were designed using MethPrimer 2.0 (Appendix A). Briefly, RiboProm1 and RiboProm2 are located in the promoter region of the rDNA unit. The former is upstream of the rDNA promoter, whereas the latter encompasses the core promoter element (CP) as well as the upstream control element (UCE) of the rDNA unit. The 18S and 28S targets are located at the 5′ end of their respective rDNA sequences (Figure 9A). RiboProm1, RiboProm2, 18S and 28S contain 37, 26, 27 and 30 CG sites, respectively (Appendix A). LINE-1 target is located in the 5′UTR of the repeat unit, close to the transcription start site (Figure 9B). The Alu target is located in the body of the repeat unit (Figure 9C). LINE-1 and Alu repeat targets, respectively, contain 18 and 12 CG sites (Appendix A). Forward and reverse primers were added at each 5′ end with Nextera^TM^ adapter sequences, respectively, TCGTCGGCAGCGTCAGATGTGTATAAGAGACAG and GTCTCGTGGGCTCGGAGATGTGTATAAGAGACAG. In addition, a random nucleotide spacer (N) was included between Illumina adapters and primers to increase sequencing variability.

Sequencing libraries were generated using a two-step PCR approach. Briefly, 5 ng of bisulfite-converted DNA was amplified using Phusion U (ThermoFisher, Waltham, MA, USA) added with 1M Betaine (Merk, Darmstadt, Germany), 150 nM forward and reverse primers, 1.75 mM MgCl_2_ (Agena Bioscience, San Diego, CA, USA) and 200 μM dNTP (ThermoFisher, Waltham, MA, USA). Thermal cycler conditions were set as follows: 1× cycle at 95 °C for 1′ 40 ″; 1× cycle at 98 °C for 1′; 1× cycle at 58 °C for 2′; 1× cycle at 72 °C for 1′; 36 cycles at 98 °C for 10″, 58 °C for 40″ and 72 °C for 20″; 1× cycle at 72 °C for 5′; and hold at 4 °C. Amplification products were pooled sample-wise and purified using MagSi-NGS plus beads (MagTivio BV, Nuth, The Netherlands). Afterward, 10 uL of pooled and purified samples were indexed using Illumina Nextera XT Index Set A (Illumina, San Diego, CA, USA) as indicated in the Nextera Library Prep Guide. Finally, indexed libraries were purified yet again and normalised to 4 nM. Sequencing was performed on an Illumina Miseq Platform with Micro v2 300PE chemistry (Illumina, San Diego, CA, USA).

### 4.5. Sequencing Data Handling

Pair-end reads were quality-checked using FastQC (https://www.bioinformatics.babraham.ac.uk/projects/fastqc/ (accessed on 25 May 2023)). Adapter sequences were trimmed off using *cutadapt* [78], and, finally, reads were merged together using *PEAR* [79], with a minimum of 20 overlapping residues and a maximum read length of 450. Assembled reads were then converted from FASTQ to FASTA using *seqtk.* All processing tools were compiled in anaconda2 environments.

### 4.6. DNA Methylation Data Extraction

To extract CpG DNAm profiles for each target sequence, the *AmpliMethProfiler* pipeline was applied [80]. This pipeline was developed to analyse deep bisulfite sequencing data. Briefly, for each sample, the *AmpliMethProfiler* tool performs a quality check on sequenced reads by removing those with mean Phred < 33 *(--perfQual*). Subsequently, it recovers all reads with at least 80% alignment identity with bisulfite-specific primers provided *(--primTresh 0.8*) and retains all reads whose length differs less than 20% from that of a reference sequence provided *(--threshLen 0.2*). Afterwards, each read is aligned to a reference sequence provided using *blastn*. Alignment parameters were modified from default in order to switch off automatic low-complexity, repetitive sequence masking *(--dust no*). This allows us to limit read loss, as bisulfite-conversion and subsequent amplification reduce sequence complexity by converting unmethylated Cytosines (C) into Thymines (T). For each aligned read, AmpliMethProfiler generates a DNAm profile by assigning binary values to CpG positions that were found methylated (1) or unmethylated (0). If a CpG position contains a nucleotide other than T or C, AmpliMethProfiler assigns an unknown (2) value. Samples with coverage lower than 400 reads were excluded from the analysis. After filtering, sequencing coverage varied between 1252 and 9432 (3858 ± 2140) for 18S, 1316 and 11,481 (5147 ± 2462) for 28S, 809 and 7512 (3027 ± 1240) for Alu, 822 and 7838 (3454 ± 1477) for LINE-1, 973 and 7580 (3995 ± 1539) for RiboProm1 and 864 and 8366 (3918 ± 1736) for RiboProm2.

Finally, average DNAm levels of each CpG unit are calculated as a percentage of methylated Cs on the total number of methylated and unmethylated Cs. 

For each cell line, average DNAm was compared between the treated and untreated groups by Student t-test, using the ANOVA function in R (v3.6).

### 4.7. Epihaplotype Analysis

AmpliMethProfiler provides information on the DNAm status of each CpG site at the single DNA molecule level. The specific combination of methylated and unmethylated CpGs on a single strand of DNA defines an epihaplotype. To study epihaplotype organisation, a pipeline of analysis was developed using AmpliMethProfiler built-in functions. As the number of possible methylated and unmethylated CpG combinations largely exceeds sequencing depth, singletons, namely epihaplotypes occurring only once in the entire sample, were removed from epihaplotype count tables (*filter_otus_from_otu_table.py –n 2*). Epihaplotype composition and distribution within each sample were then evaluated using a qiime-based pipeline implemented within the AmpliMethProfiler environment (*qiime_analysis.py*). Epihaplotype diversity was calculated through the Shannon alpha-diversity index. For each cell line, samples were grouped as either treated or control, and then observations were rarefied by randomly resampling the pool of epihaplotypes at increasing depth (from 10 to median overall coverage). Shannon indexes were calculated at a depth equal to the number of reads of the sample with the lowest coverage. However, it emerged that none of the Shannon index rarefaction curves generated was able to reach plateau at the selected depth. Therefore, in order to increase the number of reads on which observations were rarefied, we included an additional filter and removed all samples with a maximum coverage lower than 60% of the median coverage for each amplicon. Once the dataset was filtered, for every target amplicon, Shannon indexes were compared between samples exposed to radiofrequency radiation and to sham using Student’s *t*-test.

## Figures and Tables

**Figure 1 ijms-24-09380-f001:**
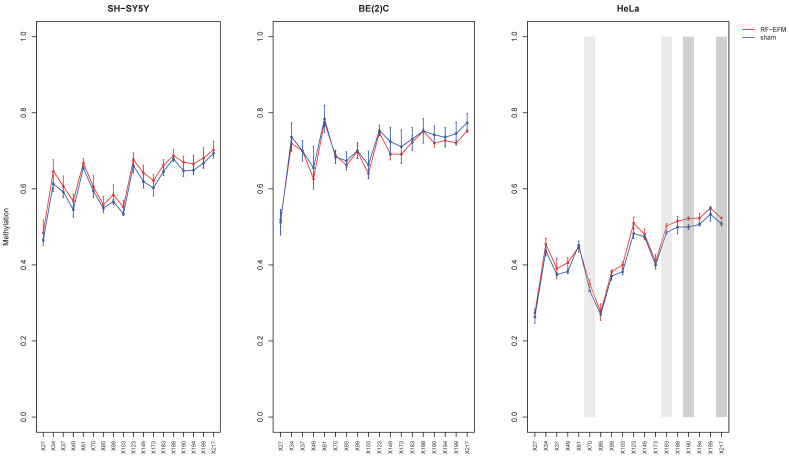
DNAm profiles of LINE-1 target in cell lines exposed to RF-EMF. Lineplot showing average DNAm ± standard deviation for every CpGs assayed in LINE-1-targeting amplicon for SH-SY5Y, BE(2)C and HeLa cell lines exposed to 900 MHz radiation (red) or to sham (blue). CpGs with *p*-value < 0.01 are highlighted with dark grey boxes; CpGs with *p*-value between 0.01 and 0.05 are highlighted with light grey boxes.

**Figure 2 ijms-24-09380-f002:**
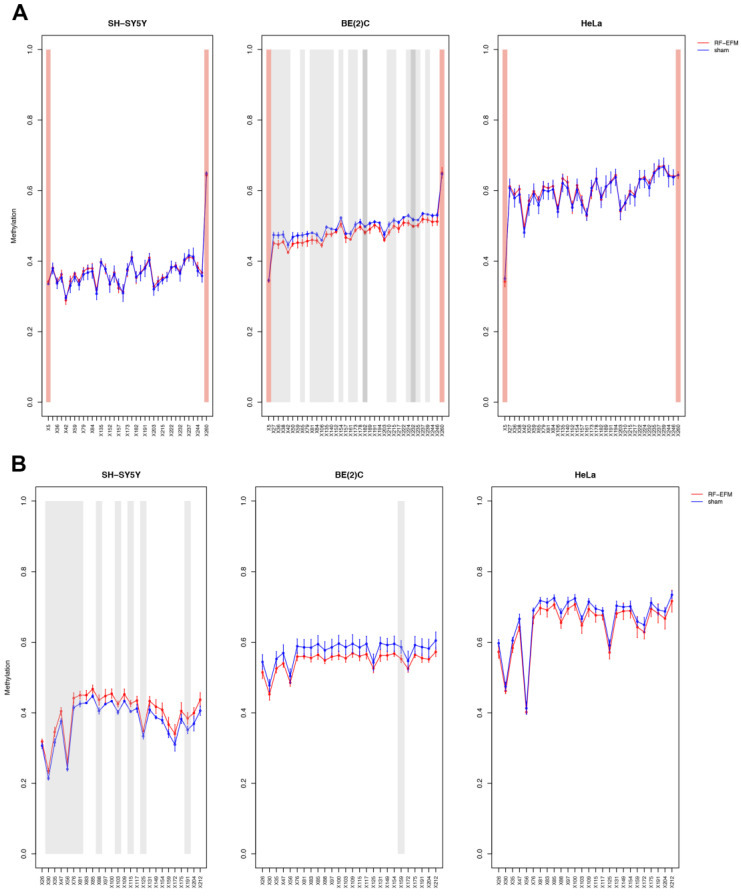
DNAm profiles of rDNA promoter targets in cell lines exposed to RF-EMF. Lineplots showing average DNAm ± standard deviation for every CpGs assayed in (**A**) RiboProm1 and (**B**) RiboProm2 amplicons for SH-SY5Y, BE(2)C and HeLa cell lines exposed to 900 MHz radiation (red) or to sham (blue). CpGs with *p*-value < 0.01 are highlighted with dark grey boxes; CpGs with *p*-value between 0.01 and 0.05 are highlighted with light grey boxes. CpGs, which were discarded due to poor calibration, are highlighted with red boxes.

**Figure 3 ijms-24-09380-f003:**
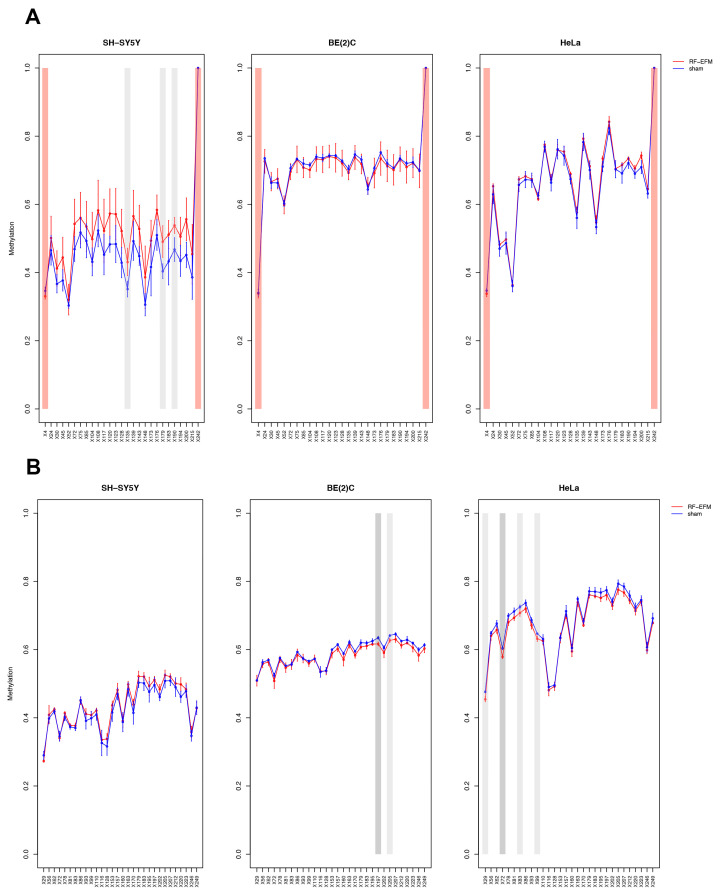
DNAm profiles of rDNA body in cell lines exposed to RF-EMF. Lineplots showing average DNAm ± standard deviation for every CpGs assayed in (**A**) 18S and (**B**) 28S amplicons for SH-SY5Y, BE(2)C and HeLa cell lines exposed to 900 MHz radiation (red) or to sham (blue). CpGs with *p*-value < 0.01 are highlighted with dark grey boxes; CpGs with *p*-value between 0.01 and 0.05 are highlighted with light grey boxes. CpGs, which were discarded due to poor calibration, are highlighted with red boxes.

**Figure 4 ijms-24-09380-f004:**
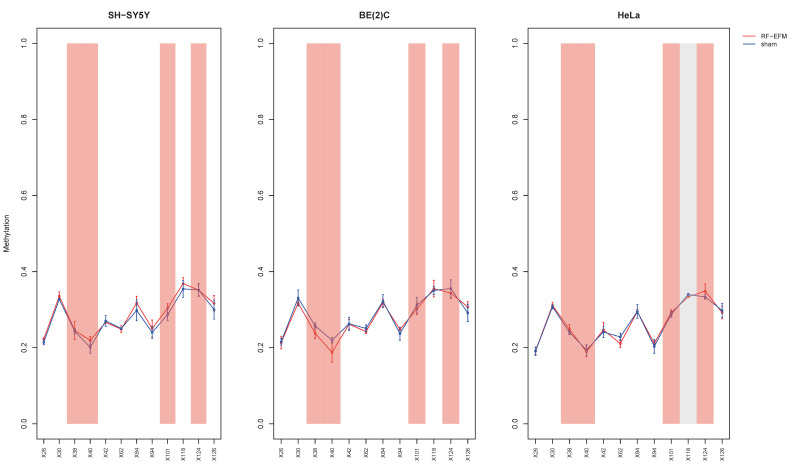
DNAm profiles of Alu repeat unit in cell lines exposed to radiofrequencies. Lineplots showing average DNAm ± standard deviation for every CpGs assayed in Alu repeat amplicon for SH-SY5Y, BE(2)C and HeLa cell lines exposed to 900 MHz radiation (red) or to sham (blue). CpGs with *p*-value between 0.01 and 0.05 are highlighted with light grey boxes. CpGs, which were discarded due to poor calibration, are highlighted with red boxes.

**Figure 5 ijms-24-09380-f005:**
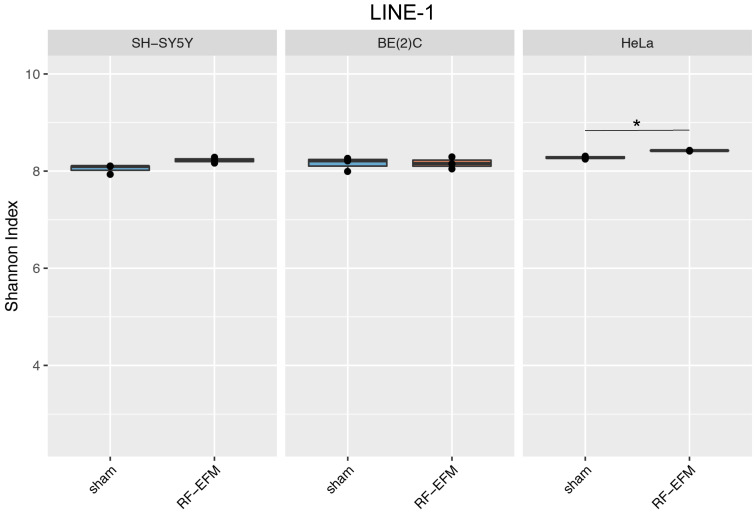
Epihaplotype diversity of DNAm profiles of LINE-1 amplicon in cell lines exposed to radiofrequencies. Boxplots reporting Shannon diversity indexes for DNAm epihaplotypes for LINE-1 amplicon for SH-SY5Y, BE(2)C and HeLa cell lines exposed to 900 MHz radiation (orange) or to sham (blue) (* *p*-value < 0.05).

**Figure 6 ijms-24-09380-f006:**
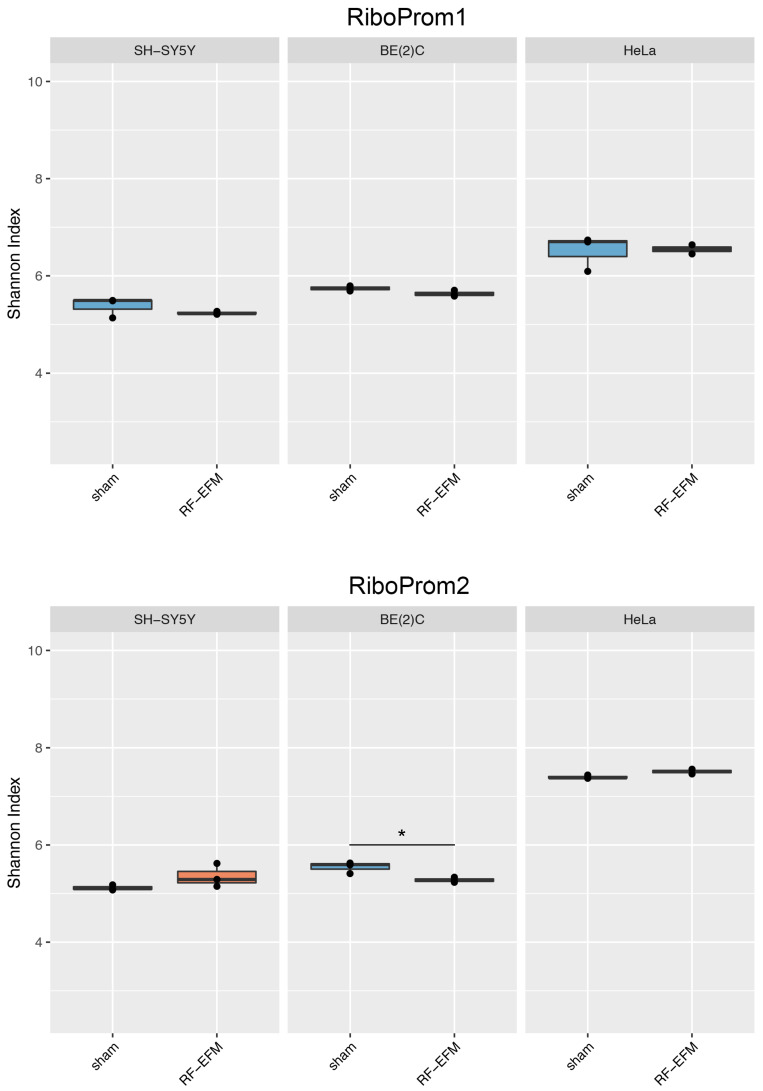
Epihaplotype diversity of DNAm profiles of rDNA promoter in cell lines exposed to RF-EMF. Boxplots reporting Shannon diversity indexes for DNAm epihaplotypes of RiboProm1 and RiboProm2 amplicons for SH-SY5Y, BE(2)C and HeLa cell lines exposed to 900 MHz radiation (orange) or to sham (blue) (* *p*-value < 0.05).

**Figure 7 ijms-24-09380-f007:**
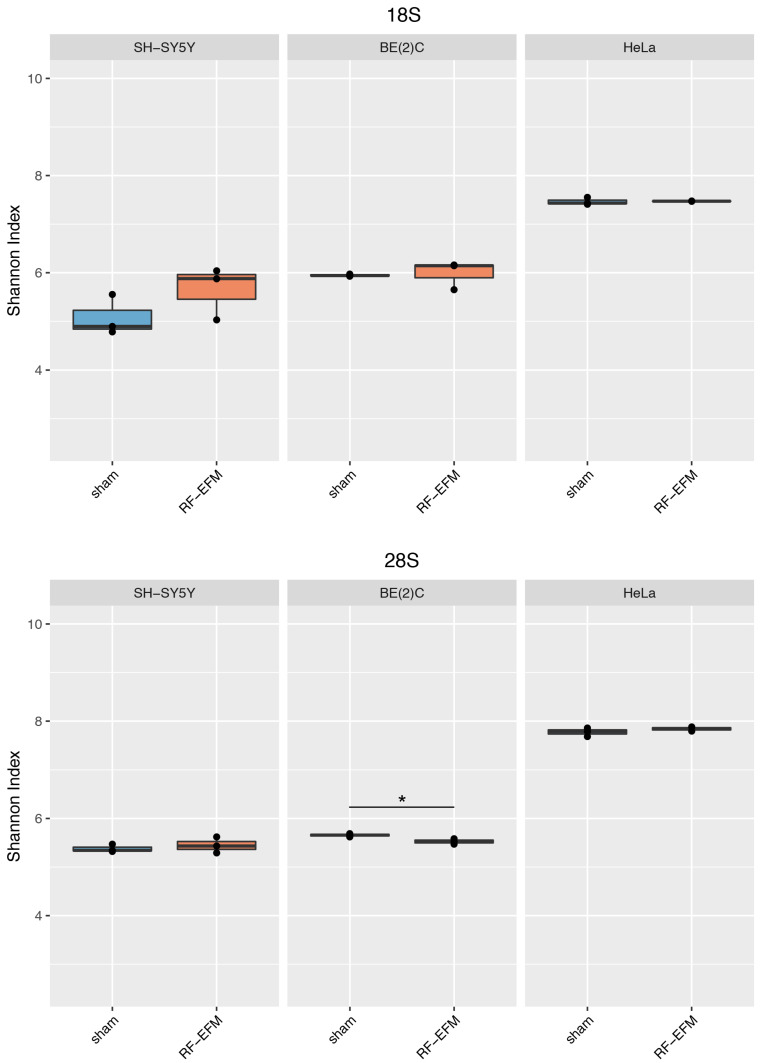
Epihaplotype diversity of DNAm profiles of rDNA body in cell lines exposed to RF-EMF. Boxplots reporting Shannon diversity indexes for DNAm epihaplotypes of 18S and 28S amplicons for SH-SY5Y, BE(2)C and HeLa cell lines exposed to 900 MHz radiation (orange) or to sham (blue) (* *p*-value < 0.05).

**Figure 8 ijms-24-09380-f008:**
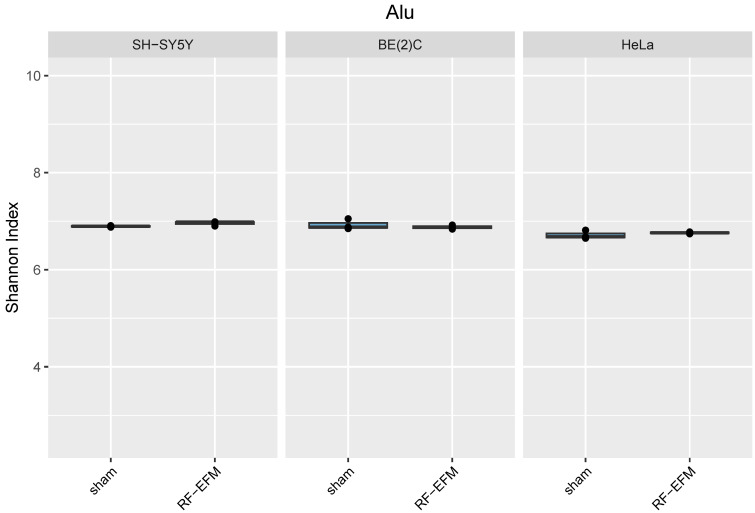
Epihaplotype diversity of DNAm profiles of Alu repeat unit in cell lines exposed to RF-EMF. Boxplots reporting Shannon diversity indexes for DNAm epihaplotypes of Alu amplicon for SH-SY5Y, BE(2)C and HeLa cell lines exposed to 900 MHz radiation (orange) or to sham (blue).

**Figure 9 ijms-24-09380-f009:**
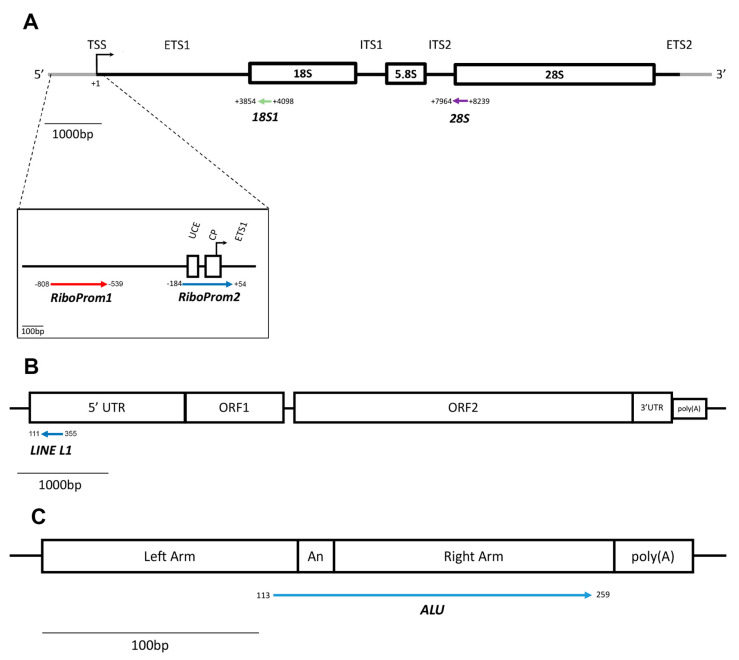
Structure of the rDNA, LINE-1 and Alu units and localisation of target regions assessed. (**A**) Structure and organisation of the rDNA repeat unit. The position of each amplicon (coloured arrows) assayed through target bisufite sequencing is defined by its distance (in base pairs) from the transcription start site (TSS) to the 45S pre-ribosomal RNA transcript (chr21:8202347–8222335, GRCh38/hg38). UCE = upstream control element; CP = core promoter; TSS = transcription start site; ETS = external transcribed spacer; ITS = internal transcribed spacer; 18S = 18S rRNA coding region; 5.8S = 5.8S rRNA coding region; 28S = 28S rRNA coding region. (**B**) Structure of LINE-1 repeat unit. The amplicon (coloured arrow) included in the targeted-bisulfite sequencing assay is located in the 5′UTR of the Human LINE-1 (L1.3) repetitive element (GenBank: L19088.1). UTR = untranslated region; ORF = open reading frame. (**C**) Structure of the Alu repeat unit. The amplicon (coloured arrow) encompasses the A-rich region and covers the right arm of the repeat unit. The orientation of the arrows representing the amplicons indicates the 5′-3′ direction of the sequenced DNA molecule. bp = base pairs.

## Data Availability

Scripts and bioinformatic tools used to generate DNAm data from bisulfite sequencing are available at https://github.com/LabBrainAgeing/AmpliMethProfiler_RepetitiveElements (accessed on 25 May 2023) Sequencing data are available online at NCBI Sequence Read Archive with the following accession number: PRJNA957105.

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
