# Peer review of "Evaluation of DNA Methylation Profiles of LINE-1, Alu and Ribosomal DNA Repeats in Human Cell Lines Exposed to Radiofrequency Radiation"

_ijms, 2023, doi:10.3390/ijms24119380_

Round 1

Reviewer 1 Report

The work titled: Evaluation of DNA methylation profiles of LINE-1, Alu and ribosomal DNA repeats in human cell lines exposed to radiofrequency radiation is suggestive and ingenious in employing highly repetitive regions with implications for instability in cancer, aging, and other relevant biological phenomena.

The experiments are well described and experimentally seem robust. However, the effects of radiofrequency radiation are too subtle to be considered relevant, although in some cases, the result was well preserved if subtle.

A consideration that the authors must attend to is the justification of the cell model used; the selection of the lines is not clear regarding the management of the repair systems, stability, and epigenetic control of DNA; a consideration was to be expected in the introduction.

On the other hand, if the hypothesis is that the effect is via ROS, a control with exposure of a ROS generator to its epigenetic targets would be expected.

Some minor issues:

On some occasions, they use Kb, and in others, kb.

They do not define “IGS” as an acronym.

In the description of Figure 1, some gray boxes are mentioned that are not shown in the figure.

The definition of Figure 4 mixes elements of Figures 4 and 3.

Reviewer 2 Report

The authors described the side effect of EM radiation on promoter hypermethylation of cancer involve genes. The study seems interesting but some comment should be replied by authors to more illustrate the research strategy.

Why the authors used the 900 MHz for treatment of cancer cells? The interpretation for the higher or lower radiation should be discussed in manuscript.

The distance of signal generator to the incubated cells should be indicated and discussed.

If this radiation emitted from regular mobile phones why the authors did not use the mobile device as the control factor.  

As the risk factor the authors should illustrate which devices generate this frequency? Do we receive these signals in our regular life? (in off or on mode of devices)

Minor editing of manuscript is required.  

Round 2

Reviewer 2 Report

After revision process, I recommend publication of manuscript.